# The role of seasonal malaria chemoprevention in the effect of azithromycin on child mortality: A secondary analysis of the CHAT cluster randomized clinical trial

Elisabeth A. Gebreegziabher[1,2], Mamadou Ouattara[3], Mamadou Bountogo[3], Boubacar Coulibaly[3], Valentin Boudo[3], Thierry Ouedraogo[3], Elodie Lebas[1], Huiyu Hu[1], Kieran S. O'Brien[1,2,4], Michelle S. Hsiang[2,5,6], David V. Glidden[2], Benjamin F. Arnold[1,4], Thomas M. Lietman[1,4], Ali Sié[3], Catherine E. Oldenburg[1,2,4]*

1 Francis I. Proctor Foundation, University of California San Francisco, San Francisco, California, United States of America, 2 Department of Epidemiology and Biostatistics, University of California, San Francisco, California, United States of America, 3 Centre de Recherche en Sante de Nouna, Nouna, Burkina Faso, 4 Department of Ophthalmology, University of California, San Francisco, California, United States of America, 5 Malaria Elimination Initiative, Institute for Global Health Sciences, University of California San Francisco, San Francisco, California, United States of America, 6 Department of Pediatrics, Division of Pediatric Infectious Diseases, San Francisco, California, United States of America

* catherine.oldenburg@ucsf.edu

## Abstract

The objective of this study was to examine whether the effect of mass Azithromycin (AZ) distribution on all-cause mortality among children under 5 varies with seasonal malaria chemoprevention (SMC) administration season or coverage. This was a secondary analysis of the Community Health with Azithromycin Trial (CHAT), a cluster-randomized, placebo-controlled trial of twice-yearly AZ treatment in 341 communities in the Nouna District, Burkina Faso. All communities received SMC as standard-of-care. SMC administration and coverage data were provided from National Malaria Control Program. SMC season was defined as the period during and following SMC (July-December) versus the no SMC season (January-June). SMC coverage was assessed as proportion of the population covered and by whether it was below or above a threshold of 80%. We used Poisson regression models with person-time at risk as an offset and robust standard error to analyze mortality rates by treatment group and SMC subgroups and assessed interaction on both multiplicative and additive scales. Mortality was higher in SMC seasons for both arms. Compared to placebo, the mortality rate in AZ clusters was 0.77 (95% CI: 0.60 to 0.98) during SMC season, while it was 0.89 (95% CI: 0.68 to 1.15) during the non-SMC seasons. In clusters with <80% SMC coverage, the effect of AZ was 0.73 95%CI (0.56 to 0.96) and in clusters with ≥80% SMC coverage, it was 1.0 95%CI (0.59 to 1.69). The interaction between AZ and SMC season or coverage was not statistically significant on the additive or multiplicative scales. While our findings did not reach

**Data availability statement:** De-identified source data for the study are available via the Open Science Framework (https://doi.org/10.17605/OSF.IO/MAUT3).

**Funding:** The CHAT trial was supported by the Gates Foundation (grant number OPP1187628 to TML and CEO). The conclusions and opinions expressed in this work are those of the authors alone and should not be attributed to the Foundation. Under the grant conditions of the Foundation, a Creative Commons Attribution 4.0 License has already been assigned to the Author Accepted Manuscript version that may arise from this submission. Research reported in this manuscript was also supported by the National Institutes of Health Eunice Kennedy Shriver National Institute of Child Health & Human Development (NIH/NICHD) F31 Award (1F31HD114434-01A1 to EAG.). The funders had no role in study design, data collection and analysis, decision to publish, or preparation of the manuscript.

**Competing interests:** The authors have declared that no competing interests exist.

statistical significance, they raise the question of whether prioritizing MDA AZ during high transmission periods or in regions with low SMC coverage could be beneficial. Further research is needed to determine if targeting these periods or areas could further reduce child mortality.

## Introduction

Many regions of sub-Saharan Africa have high child mortality rates [1]. Infectious diseases, in particular acute diarrheal and respiratory infections, and malaria, are the leading causes of illness and death in children in low- and middle-income countries [2]. Antimicrobial-based strategies such as mass drug administration of azithromycin (MDA AZ) and seasonal malaria chemoprevention (SMC) have been used to reduce child mortality in these settings [3,4]. Azithromycin (AZ) has been shown to reduce all-cause mortality as well as infectious morbidity due to diarrhea, pneumonia, and malaria [3,5,6]. The MORDOR study which randomized communities in Malawi, Niger, and Tanzania to four twice-yearly mass distributions of either oral AZ or placebo found that childhood mortality was lower in communities randomized to AZ, with the largest effect (a relative reduction in all-cause mortality of 18.1%) seen in Niger, where SMC was not being distributed [3]. The Community Health with Azithromycin Trial (CHAT) completed in 2023 and the AVENIR (Azithromycine pour la Vie des Enfants au Niger: Implementation et Recherche) trial in Niger also demonstrated that mass distribution of AZ remains effective at reducing child mortality in a setting receiving SMC [7,8]. The finding of these trials was consistent with MORDOR-Niger, showing a reduction in all-cause mortality overall in communities randomized to AZ compared to placebo [7].

SMC involves monthly administration of a 3-day regimen of sulfadoxine-pyrimethamine and amodiaquine (SP-AQ) during the malaria season in regions with highly seasonal transmission, where there is no evidence of resistance to these drugs [9]. SMC treats current infection and provides protection against future infections for up to 3–4 weeks [10]. As of 2020, following the WHO policy recommendation, 13 countries in Africa have adopted SMC with different scales of implementation [11,12]. Large-scale evaluations have found SMC effective at reducing malaria infection and all-cause mortality [13,14]. Despite the adoption of SMC, it is not clear how the antimalarial, antibacterial, anti-inflammatory and overall benefits of SP in SMC might affect the benefit of AZ for child mortality [15]. Since Sulfadoxine is a broad-spectrum, long acting antibiotic, it may influence infections that AZ is effective against. Previous evidence for the combined effect of AZ and SP in pregnant women showed that the combination of these drugs was superior in antimalarial activity compared to SP with chloroquine [16,17] and SP alone [18]. A recent RCT found that children receiving the addition of AZ to SMC, versus placebo and SMC, had fewer episodes of gastrointestinal infections, upper respiratory tract infections, and non-malarial febrile illnesses. However, there was no significant difference in incidence of death or hospitalization between these groups [19]. The lack of benefit of AZ for

mortality observed in this trial raises questions about whether the mortality benefit of AZ may be lower when SMC is also administered. On a larger scale, these questions are relevant for the implementation of MDA AZ, particularly in relation to the timing of AZ administration (since SMC is administered seasonally) and for identifying communities that would benefit most based on SMC coverage. Communities with gaps in SMC coverage may also have other vulnerabilities and gaps in care, for which AZ might be particularly effective [20].

The potential heterogeneity in the effect of AZ calls for further studies before its widespread use can be recommended [21]. Studies are needed to refine key aspects of implementation, such as the optimal dose, frequency, timing [22], and the populations to target or prioritize [23,24]. Additionally, understanding how malaria seasonality and the concurrent use of malaria interventions like SMC impact the effectiveness of MDA AZ programs can help in tailoring these interventions to enhance their benefits, use resources efficiently and develop integrated strategies.

Using data from the CHAT cluster-randomized controlled trial in Burkina Faso [7] and the country's National Malaria Control Program (NMCP) data on SMC administration and coverage, we examined whether the effect of mass AZ distribution on mortality among children aged 1–59 months varies with SMC administration season or SMC coverage. We hypothesized that the effect of AZ will vary with SMC season and coverage level.

## Methods

### Ethics statement

The CHAT trial which served as basis for these analyses was reviewed and approved by the Institutional Review Boards at the University of California, San Francisco (IRB Number: 17–24230) and the Comité National d'Ethique pour la Recherche (National Ethics Committee of Burkina Faso (IRB Number: 2018-8-111) in Ouagadougou, Burkina Faso. Written informed consent was obtained from the caregiver of each participant.

### Study design, setting, population and data collection

This was a post hoc secondary analysis of the CHAT trial (ClinicalTrials.gov, NCT03676764) and additionally included SMC data from NMCP. The methods and primary results have been reported in detail previously [7,25]. Complete methods for the trial are also shown in the trial protocol (S1 Text). In brief, CHAT was a cluster-randomized placebo-controlled trial designed to evaluate the efficacy of mass AZ strategies [25]. The trial was conducted from August 2019 to February 2023 and enrolled children aged 1–59 months in Nouna District of Burkina Faso. It covered both the Nouna Health and Demographic Surveillance Site (one third of Nouna District) [26], as well as the non-HDSS area in Nouna [25]. A map of the study area is included in the primary analysis of the trial [7]. Communities were randomized in a 1:1 ratio to receive twice yearly distribution of a single oral dose of azithromycin or placebo. Children in these communities received their respective treatments for 3 years (from August 2019- February 2023). The overall coverage of MDA was approximately 90% for both AZ and placebo arms. Every 6 months, a census was conducted documenting births, deaths, pregnancies, and migration. CHAT had an open cohort design with children contributing different amounts of person time, and vital status recorded at each 6-month census (phases) for each child enrolled. Over the study period, 1,086 deaths over 119,139 person-years were observed (Fig 1) [7]. Six rounds of census and treatment occurred during the study corresponding to the following phases: 0–6 months, 6–12, 12–18, 18–24, 24–30 and 30–36 months (Fig 2). Trial data were aggregated by cluster and these phases. All communities included in trial were included in these analyses.

Four monthly cycles of SMC were administered per local standard-of-care to children under 5 during the malaria transmission season from July through October each year. SMC was administered to the study area over a 3–4 day period [27]. NMCP records were reviewed to obtain SMC coverage data, including number of children treated during each cycle and the timing of treatment at the health post (CSPS) level. Each community falls within a CSPS catchment area and was assigned the SMC coverage level of its corresponding CSPS.

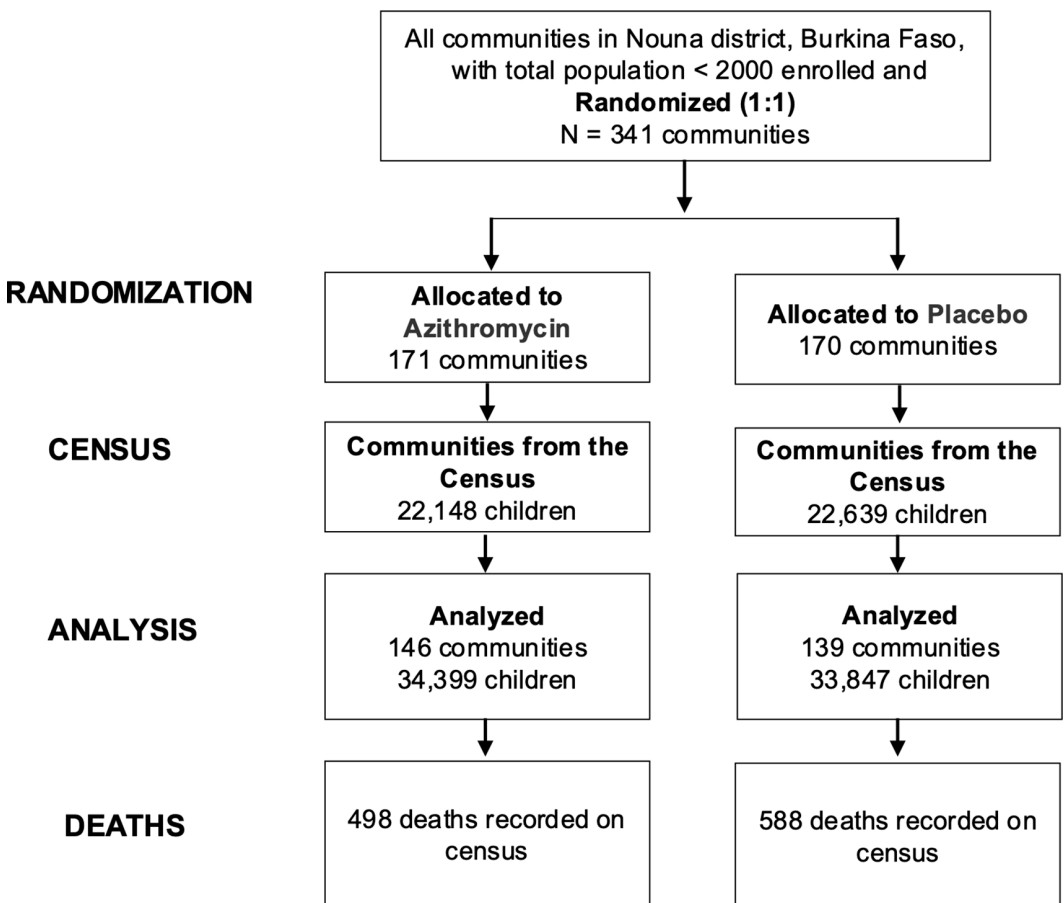

**Fig 1. Cluster and Participant Flow Diagram.**

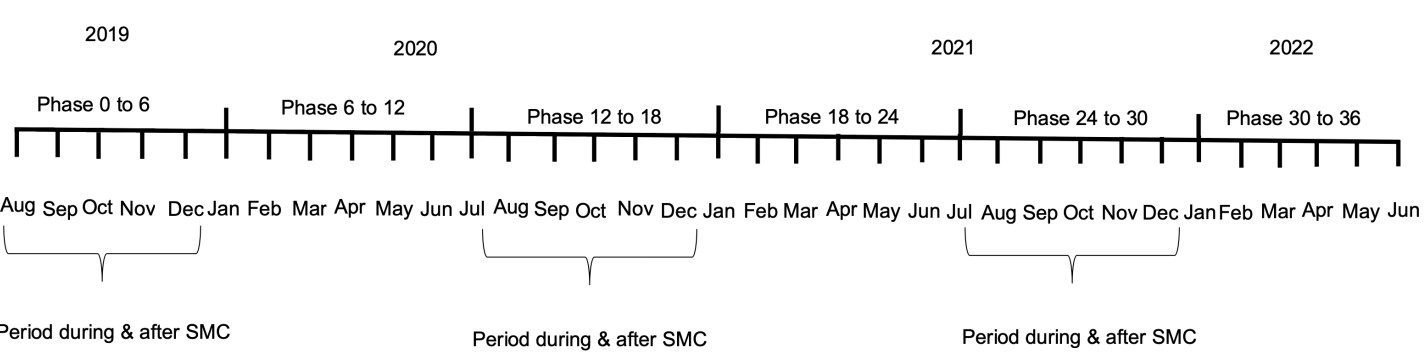

**Note**: AZ treatment and census occurred during each phase (biannual). There were four rounds of SMC treatment (July to October, one each month) each year

**Fig 2. Timeline of census and AZ treatment phases in CHAT trial and SMC seasons.**

## Statistical analysis

The primary outcome for these analyses was all-cause mortality, assessed using a biannual census. The sample size was based on childhood mortality for the overall AZ versus placebo comparison in CHAT trial. Details of sample size calculation have been reported previously [7]. For the subgroup analysis, we calculated the minimum detectable effect (additive interaction effect per 1000 children) based on methods described by Hayes & Moulton for calculating power and sample size for examining interaction in cluster randomized trials [28], This method accounts for clustering by using a randomization -unit level summary statistic [28]. Using parameters from CHAT data at 24 months, with 139 clusters per arm, 27 person-years of observation per cluster per subgroup, cluster standard deviation of 0.0223 and alpha of 0.05 for 2-sided test, we estimated that we would have 80% power to detect a delta of 1.16 (additive interaction effect of ~1 death per 1000 children). We conducted three analyses in which we assessed the effect of AZ (AZ vs placebo) in 1) the months of no SMC (January- June) versus months during and immediately following SMC (July-December), 2) by SMC coverage level (0–100%) on a continuous scale, and 3) by SMC coverage of below or above a threshold level of 80%, which was chosen a priori based on recommended WHO coverage levels. Descriptive analyses were used to report the mean number of children in each community, cluster-level mean percentages of children by age-group and sex, as well as SMC coverage levels. We used the interaction directed acyclic graph [29] shown in Fig 3 to visualize the relationship between SMC and effect of AZ on under 5 mortality examined in the interaction analyses.

For the analysis by season, the six phases were categorized as non-SMC (January-June) versus the SMC (July-December) seasons of the year, based on the timing of treatment and census. A Poisson regression model was used to analyze data at the cluster phase level with person-years included as an offset and robust standard error clustered at the cluster level to account for repeating phases. This was used to estimate the mortality rate in AZ and placebo groups, their rate ratio by season, and their interaction on the multiplicative scale. The margins command in Stata was used to estimate incidence rate differences (IRDs) with 95% CIs using the delta method. Number needed to treat to avert one death was calculated using the rate differences. Relative excess risk due to interaction (RERI) with bootstrap 95% confidence limits and 1000 repetitions was used to assess interaction on the additive scale. We evaluated interaction on both the additive and multiplicative scales to provide a comprehensive assessment of effect modification. The additive scale indicates whether the effect of the intervention is greater in one subpopulation or season than in another, making it useful for targeting specific populations or timing and for resource allocation. In contrast, multiplicative interactions provide insight into differences across risk groups on a relative scale [30].

For the analysis by coverage, we updated the SMC coverage for 2020 and 2021 originally reported by NMCP using 2006 denominators (estimates of total number of children), by the corresponding census denominators from CHAT to

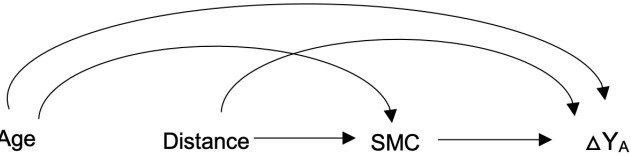

**Note:** Age-age of children, Distance-distance to health facility, SMC-SMC coverage level, Y-under-five mortality, A- AZ treatment, $\Delta Y_A$-effect of AZ on child mortality. Fig 3 shows how SMC may modify the effect of AZ on child mortality, with potential confounders including the interaction of distance to a health facility with AZ, and the interaction of child age with AZ on child mortality. Adjusted analyses are presented in S3 Fig.

**Fig 3. Interaction Directed Acyclic Graphs Illustrating the Interaction Between Mass AZ Treatment and SMC.**

obtain more accurate coverage estimates for each CSPS. The SMC coverage for each cycle in each year for each CSPS was calculated as the number of children treated (obtained from NMCP) divided by the total number of children in the CSPS catchment area.

SMC coverage percentages for each cluster and each year (2019, 2020, 2021) were linked to the trial data by cluster. Since SMC coverage estimates for 2019 were not available and given the high correlation (>90%) in CSPS level coverage between 2020 and 2021, coverage estimates for 2020 were used for both 2019 and 2020. For the coverage estimate for each year, we used the average coverage level across the 4 cycles of SMC distributed during that year. Clusters with coverage estimates above 100% (~8%) were capped to 100%. SMC coverage was scaled for each unit increase to represent 10% increase in coverage. Given the short duration of protection of SMC [31], for the analysis by coverage, we restricted the data to the second half of each year (July-December), the period during and immediately after SMC was administered. The analysis dataset included the treatment type (AZ vs placebo), number of deaths and person-time aggregated by cluster and phase year, along with the coverage estimate for each cluster and phase, in a longitudinal format.

For the analysis by coverage level of 80% or above, we used the dataset for the coverage analysis described above and dichotomized coverage level as below or 80% and above and examined whether the effect of AZ varied in these subgroups. We considered this cutoff reasonable since it has been noted that 69% of children receiving all SMC cycles or 90% receiving at least one cycle, can be considered high coverage [32]. For both analyses by coverage level, we used Poisson regression models with robust standard errors and similar methods described above for examining interactions on additive and multiplicative scales.

Due to the imbalance in SMC coverage between arms, we conducted sensitivity analysis adjusting for SMC coverage in the analysis by SMC season. Furthermore, we categorized the year into quarters, in a sensitivity analysis, to evaluate during which time of the year AZ distribution yielded the greatest benefit against mortality. Lastly, we examined the role of SMC coverage in the effect of AZ on child mortality, adjusting for age of children and distance to health facility, and their interactions with AZ [29], as both may influence the level of SMC coverage and affect the benefit of AZ for mortality [3,20,33]. As an additional sensitivity analysis, we compared the effect of AZ on mortality between communities in the lowest quintile (bottom 20%) and highest quintile (top 20%) of SMC coverage, providing a more pronounced contrast of extremes in coverage. This analysis further evaluated whether the benefit of AZ was greater in areas with limited SMC coverage.

We plotted the mortality rate and rate differences by SMC season, coverage level, and by a threshold coverage level of 80% to visualize changes in the effect of AZ, and in the mortality rates by arm. SAS 9.4 (SAS Institute, Cary, NC) was used for data cleaning and for generating the datasets for analyses. Stata version 14.2 (StataCorp, College Station, TX) was used for all other analyses.

## Results

The CHAT trial involved 341 communities. Of these, 171 were randomized to receive Azithromycin, with 146 being analyzed, while 170 communities were randomized to receive placebo, with 139 being analyzed (Fig 1). Characteristics of communities included in the trial are presented in Table 1. Around 30% of the children in each arm were aged 12–23 months. The age and sex distributions were similar across arms. The SMC coverage estimates that were updated by recent denominators showed variability, with an overall average coverage of 79% (SD 0.59). The average SMC coverage across rounds was slightly higher in treatment versus placebo arm, 82% versus 73%, respectively. In both arms, SMC coverage level was slightly higher in the 4th cycle compared to other cycles. In the CHAT trial, the mortality rate in the azithromycin group was 8.2 deaths per 1000 person-years while in the placebo group, it was 10.0 deaths per 1000 person-years [7].

For both arms, mortality rate was higher in the SMC seasons: 7.9 per 1000 person-years, 95% CI (6.9 to 9.0) in the first half of the year, compared to 10.3 per 1000 person-years, 95%CI (9.0 to 11.6) in the second half of the year (Fig 4A)

**Table 1. Characteristics of communities in CHAT trial.**

| | Azithromycin group (146 communities) | Placebo group (139 communities) |
|---|---|---|
| Number of Children | 34,399 | 33,847 |
| Age groups (mean %, SD) | | |
| 1–11 months | 17.7% (3.8) | 17.1% (3.6) |
| 12–23 months | 30.6% (4.3) | 30.4% (4.4) |
| 24–35 months | 15.0% (2.7) | 15.2% (3.1) |
| 36–47 months | 14.0% (2.7) | 14.1% (2.9) |
| 48–59 months | 14.0% (3.2) | 14.8% (3.8) |
| Male sex (mean %, SD) | 50.2% (4.2) | 50.0% (4.0) |
| SMC coverage (mean %, SD) | 82% (0.66) | 73% (0.49) |
| Cycle 1 (July) | 80% (0.65) | 71% (0.47) |
| Cycle 2 (August) | 80% (0.66) | 72% (0.5) |
| Cycle 3 (September) | 83% (0.66) | 74% (0.5) |
| Cycle 4 (October) | 84% (0.69) | 75% (0.51) |

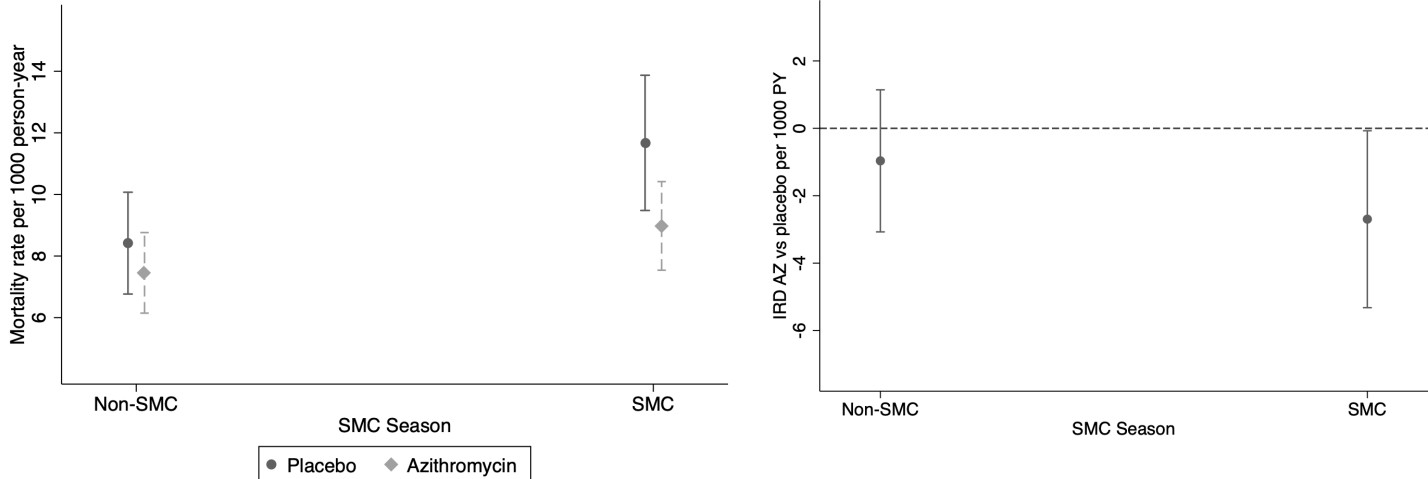

**Note:** IRD- Incidence rate difference. Interaction coefficients - multiplicative: 0.87 (0.65 to 1.16), p= 0.335, P = 0.301; additive: -0.21 (-0.57 to 0.16), P=0.268.

**Fig 4. Mortality rates and rate differences by AZ treatment and SMC season. A) Mortality rate by treatment and SMC season and B) Mortality rate difference (AZ vs placebo) by SMC season.**

In both SMC and non-SMC seasons, mortality rate was higher in placebo arm (Table 2). The interaction between AZ and SMC season was not statistically significant on either the multiplicative or additive scale. The effect of AZ compared to placebo was 0.77, 95%CI (0.6 to 0.98) in SMC season vs 0.89 (0.68 to 1.15) in non-SMC season (Fig 4B).

Fig 5 shows the mortality rates by arm and the effect of AZ at different SMC coverage levels. The interaction between AZ and SMC coverage (both on continuous scale and dichotomous at 80% coverage threshold) was not statistically significant on either the multiplicative or additive scale (P>0.05). However, the benefit of AZ appeared to slightly lower with increasing SMC coverage (Fig 5A and 5B). The effect of AZ compared to placebo in clusters with <80% SMC coverage was 0.73 95%CI (0.56 to 0.96) and in clusters with ≥80% SMC coverage, it was 1.0 95%CI (0.59 to 1.69). These are also shown in S1 Table.

**Table 2. Effect of Azithromycin vs Placebo Distribution on Child Mortality overall and by SMC season.**

| | Overall (Jan-Dec) | Non-SMC (Jan-Jun) | During/Post-SMC (Jul-Dec) |
|---|---|---|---|
| Mortality rate per 1000 PY (all clusters) | 9.1 (8.1 to 10.1) | 7.9 (6.9 to 9) | 10.3 (9 to 11.6) |
| Mortality rate per 1000 PY in AZ | 8.2 (7.1 to 9.4) | 7.5 (6.1 to 8.8) | 9 (7.5 to 10.4) |
| Mortality rate per 1000 PY in Placebo | 10.0 (8.4 to 11.7) | 8.4 (6.8 to 10.1) | 11.7 (9.5 to 13.9) |
| IRR (AZ vs placebo) | 0.82 (0.66 to 1.01) | 0.89 (0.68 to 1.15) | 0.77 (0.6 to 0.98) |
| IRD (AZ vs placebo) | -1.82 (-3.8 to 0.14) | -0.97 (-3.07 to 1.14) | -2.7 (-5.32 to -0.07) |
| Number needed to treat to prevent one death | 549 | 1037 | 371 |
| Interaction Coeff multiplicative | 0.87 (0.65 to 1.16), p = 0.335 | | |
| Interaction Coeff additive | -0.21 (-0.57 to 0.16), P = 0.268 | | |

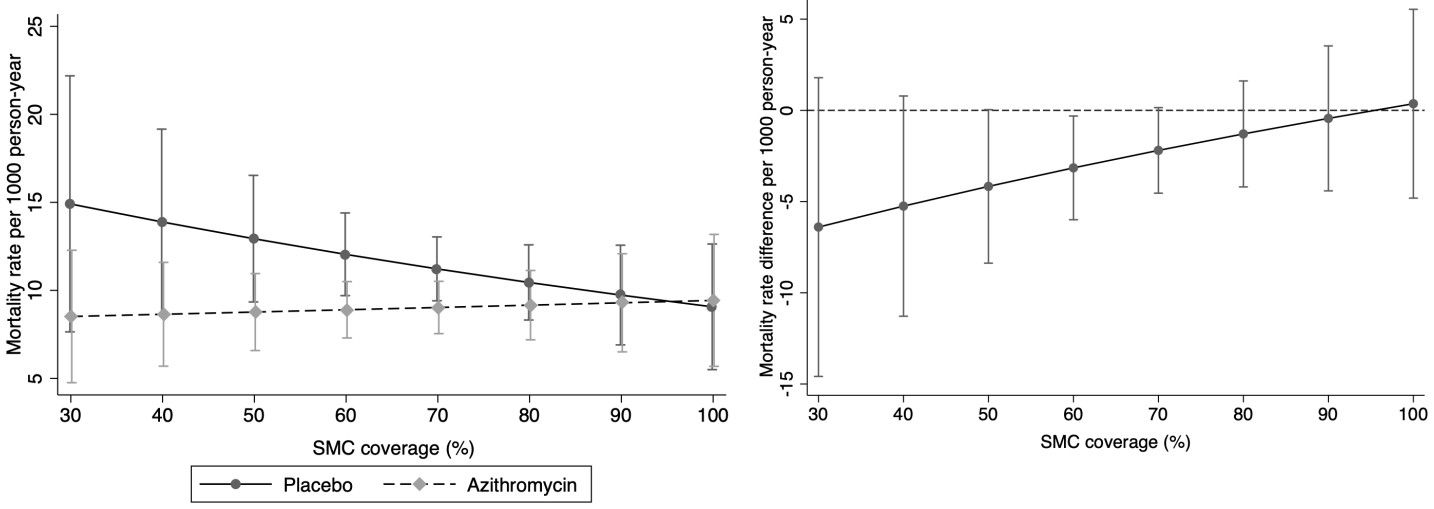

**Note:** This analysis was restricted to months during and after SMC distribution (July-December). The lowest SMC coverage was 30%. Interaction coefficients - multiplicative: 1.1 (0.9 to 1.3), P=0.297, P = 0.301; additive: 0.08 (-0.06 to 0.21), P= 0.264.

**Fig 5. Mortality rates and rate differences by AZ treatment and SMC coverage. A) Mortality rate by treatment arm at each level of SMC coverage and B) difference in mortality rate (AZ vs Placebo) by SMC coverage.**

Although the interaction was not statistically significant, the mortality rate was lower in the AZ arm compared to the placebo arm (IRR AZ vs placebo = 0.73, 95%CI (0.56 to 0.96) in clusters with SMC coverage below 80%. No significant difference in the mortality rate was observed between the AZ and placebo arms in clusters with SMC coverage of 80% or above: IRR = 1.0, 95%CI (0.59 to 1.69), Fig 6A and 6B, S2 Table).

In the sensitivity analysis, we found that adjusting for coverage in the analysis by season generated similar findings in which the effect of AZ compared to placebo was 0.77 95%CI (0.6 to 0.98) in SMC season vs 0.89 95%CI (0.68 to 1.15) in non-SMC season (S3 Table, S1 Fig).

In the sensitivity analysis by quarters, the interaction between AZ and quarter did not reach statistical significance (P=0.203). However, the benefit of AZ appeared to be more pronounced when administered in the last 3 months of the year (October–December); IRD = -4.0 per 1,000 person-years (95% CI: -7.9 to -0.11) (S2 Fig). In the analysis in which we examined the role of SMC coverage in the effect of AZ on child mortality, adjusting for age of children and distance to

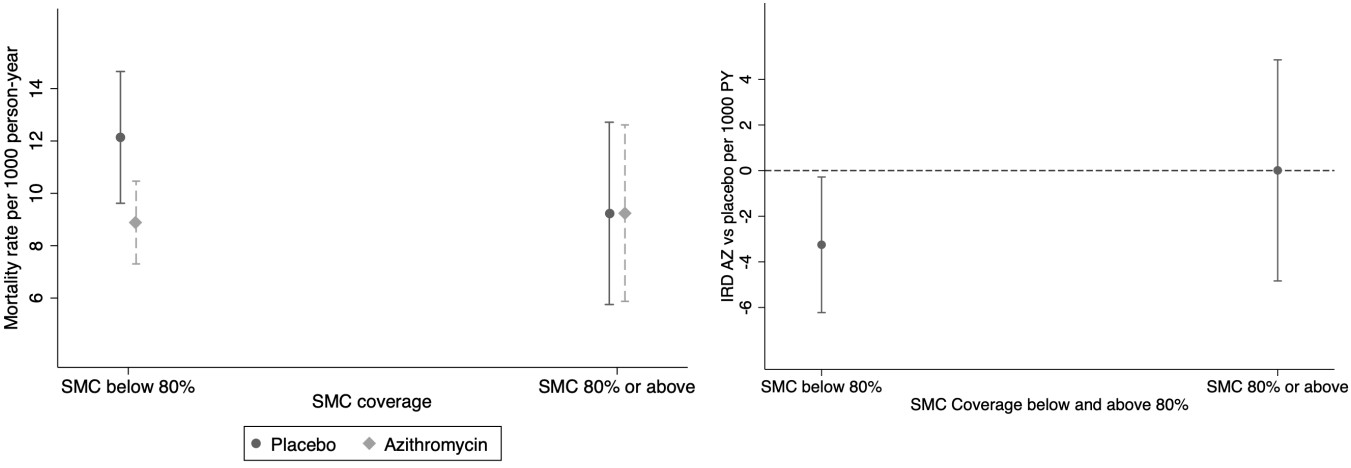

**Note:** IRD = Incidence Rate Difference. Interaction coefficients - multiplicative: 1.37 (95% CI: 0.76 to 2.48), P = 0.301; additive: 0.27 (95% CI: –0.19 to 0.72), P = 0.247.

**Fig 6. Mortality rates and rate differences by AZ treatment and SMC coverage threshold of 80%. A) Mortality rate by treatment and SMC coverage and B) Mortality rate difference (AZ vs placebo) by SMC coverage threshold of 80%.**

health facility and their interaction with AZ, we found that the results were similar in which we did not reach statistical significance but the benefit of AZ appeared to slightly lower with increasing SMC coverage (S3 Fig). In the additional sensitivity analysis, comparing communities in the lowest (bottom 20%, with SMC coverage ranging from 30–52%) and highest (top 20%, with coverage ranging from 90–100%) quintiles of SMC coverage, we observed a contrast in effect. Although statistical significance was not reached, the results again suggested that the benefit of AZ may have been slightly higher in areas with lower SMC coverage (S4 Fig).

## Discussion

We evaluated how the effect of AZ changes by seasons of SMC administration and coverage. We found that the interaction between AZ and SMC season or coverage did not reach statistical significance on both the additive or multiplicative scales. Mortality rates appeared higher during SMC seasons for both AZ and placebo-treated clusters. Although not significant, the benefit of AZ may have been higher during these high-mortality SMC seasons. Mortality rates appeared to decrease with increasing SMC coverage in placebo clusters but not in AZ clusters. Compared to placebo, our results suggested that AZ may have had an increasing benefit as SMC coverage decreased. Its effect was also more pronounced in clusters with <80% versus ≥80% SMC coverage.

The effect of AZ appeared higher during and after the months when SMC was administered, particularly towards the end of the year. This is likely due to the high malaria transmission and associated high child mortality during this period [34]. Since AZ may provide short-term infection prevention and treatment by clearing infections [5,35], its impact may be higher when administered around peak morbidity and mortality periods. It is suggested that optimal administration of AZ requires targeting asynchronous peaks in morbidity and mortality from various causes, as AZ is not specific to a single pathogen [35]. Although other infectious diseases treated by AZ, such as pneumonia and diarrhea, may follow different seasonal patterns than malaria [36,37], AZ appeared to have a larger effect during this period of elevated malaria transmission and mortality. Previous studies in this region have found that malaria rates remained high in November and December, extending beyond the final cycle of SMC in October [38]. Furthermore, the sensitivity analysis (S2 Fig) suggests that the effect of AZ may be more pronounced during these late months of the year, compared to the third quarter

when SMC is primarily administered. While the overall benefit of AZ appeared greater in the post-SMC period (July–December), its enhanced effect may have been concentrated in the final quarter, when malaria and potentially mortality rates remain high without further cycles of SMC. This likely higher benefit however may also be due to a higher number of deaths to be prevented or greater statistical power to detect an effect of AZ, given the higher mortality during this season. Given concerns about the frequency of AZ in relation to resistance [21,22], as well as its short-term effects, future studies should examine whether fewer rounds of AZ, but administered during the malaria transmission and SMC season, particularly when mortality is elevated, could be beneficial.

Although the interaction was not statistically significant, the slightly higher effect of AZ in clusters with lower SMC coverage may suggest that the benefit of AZ may be greater in areas with gaps in SMC coverage. A likely contributing factor could be that SMC coverage may be related to engagement with the healthcare system, with low-coverage areas likely facing obstacles to care [33]. AZ MDA has been found more effective for children who often have less access to preventive and curative care [20]. Therefore, the lower SMC coverage may reflect broader gaps in community and facility resources related to SMC administration. Previous studies have identified several barriers to SMC uptake, including local shortages, challenges reaching mobile populations and underserved areas, and issues related to community distributors' working conditions, such as workload and incentives [33,39]. Therefore regions with lower SMC coverage may also have other gaps in routine child health interventions and care, where AZ may be particularly helpful. Although it did not reach statistical significance, the higher benefit of AZ in low SMC communities suggests that prioritizing AZ treatment in areas with inadequate SMC coverage might be beneficial.

The slightly reduced benefit of AZ in areas with high SMC coverage may also partly be due to reduction in malaria disease burden following SMC administration [27]. In regions with lower SMC coverage, important gaps in malaria prevention remain [32], making AZ MDA likely beneficial as it may have some efficacy against both malaria and bacterial infections. Although the effect of AZ on malaria infections has been inconsistent [40], analysis of causes of death from the MORDOR trial found that malaria was one of the primary causes of mortality against which AZ was beneficial [6]. With effective malaria interventions like SMC, the impact of AZ may be less apparent [41]. This is also consistent with the lack of observed benefit of AZ for mortality in the household randomized trial, where all enrolled children received SMC [19].

This study has some limitations. First, it may have been underpowered to a detect significant interaction between AZ and SMC on the mortality outcome. For the analysis by coverage level, the data was further restricted to SMC seasons, which further reduced our sample size. However, we still observed patterns suggestive of a differential effect of AZ by season and SMC coverage. Considering the lack of statistical power, is it possible that the slight differences we observed may be due to chance. These findings should be confirmed in larger studies and examined in different settings. Second, the SMC data used in our analysis were programmatic data, not prospectively collected for research purposes and therefore had less specificity and details such as daily doses or whether treatment was directly observed. Additionally, the data were provided at the health post level, and any differences in coverage levels between clusters were not captured. The absence of data for 2019 is also another limitation in our analysis. Furthermore, although we adjusted for distance to facility and age group of children, there may be other unknown factors that can affect SMC coverage as well as the effect of AZ on child mortality. However, since the goal of this these analyses was to identify subgroups that potentially benefitted the most from AZ treatment, only confounding of the primary exposure-outcome (AZ-mortality) relationship needs to be taken into account [42]. We would not expect confounding to be an issue for the AZ-mortality effect due to the randomized nature of the intervention. There may also be measurement error in our coverage level estimates of SMC. To minimize this, we updated denominators using recent census estimates of the number of children and capped outliers. Although no results reached statistical significance, the potential for Type I error remains due to multiple comparisons inherent in exploratory analyses. Lastly, since the CHAT trial included rural communities, the generalizability of these findings may be limited to such settings.

## Conclusion

Although the interactions did not reach statistical significance, our findings raise the question of whether MDA AZ programs could be more effective if prioritized during the latter part of the year, when malaria transmission and mortality are elevated. Additionally, regions with lower SMC coverage may represent a useful target for AZ MDA interventions. Future studies should explore whether prioritizing AZ administration in these high-risk periods or areas with gaps in SMC coverage leads to greater reductions in child mortality.

## Supporting information

**S1 Text. CHAT Trial Manual of Operations.**
(PDF)

**S1 Fig. A) Mortality rate by treatment and SMC season and B) Mortality rate difference (AZ vs Placebo) by season adjusting for SMC coverage level.**
(TIFF)

**S2 Fig. Mortality rate difference (AZ vs placebo) by quarter of year.**
(TIFF)

**S3 Fig. Mortality Rate by Treatment and SMC Coverage: A) Continuous Scale and C) Threshold Level; Mortality Rate Difference (AZ vs. Placebo) by SMC Coverage: B) Continuous Scale and D) Threshold Level, Adjusting for Distance to Facility, Age of Children, and Their Interaction.**
(TIFF)

**S4 Fig. Sensitivity Analysis of Mortality Rate Difference (AZ vs. Placebo) by SMC Coverage Quintiles: a) Effect of AZ across all SMC coverage quintiles; b) Comparison of the effect of AZ between communities in the lowest (bottom 20%) and highest (top 20%) SMC coverage quintiles.**
(TIFF)

**S1 Table. Effect of Azithromycin vs Placebo Distribution on Child Mortality by SMC coverage level.**
(DOCX)

**S2 Table. Effect of Azithromycin vs Placebo Distribution on Child Mortality by SMC coverage threshold of 80%.**
(DOCX)

**S3 Table. Effect of Azithromycin vs. Placebo on Child Mortality by Season, Adjusting for SMC Coverage.**
(DOCX)

## Author contributions

**Conceptualization:** Elisabeth A. Gebreegziabher, Michelle S. Hsiang, Benjamin F. Arnold, Thomas M. Lietman, Ali Sié, Catherine E. Oldenburg.

**Data curation:** Huiyu Hu, Elisabeth A. Gebreegziabher.

**Formal analysis:** Elisabeth A. Gebreegziabher, David V. Glidden, Benjamin F. Arnold.

**Funding acquisition:** Thomas M. Lietman, Catherine E. Oldenburg, Elisabeth A. Gebreegziabher.

**Investigation:** Elisabeth A. Gebreegziabher, Mamadou Ouattara, Mamadou Bountogo, Boubacar Coulibaly, Valentin Boudo, Thierry Ouedraogo, Elodie Lebas, Huiyu Hu, Kieran S. O'Brien, Thomas M. Lietman, Ali Sié, Catherine E. Oldenburg.

**Methodology:** Elisabeth A. Gebreegziabher, Kieran S. O'Brien, Michelle S. Hsiang, David V. Glidden, Benjamin F. Arnold, Thomas M. Lietman, Catherine E. Oldenburg.

**Project administration:** Mamadou Ouattara, Mamadou Bountogo, Boubacar Coulibaly, Valentin Boudo, Thierry Ouedraogo, Elodie Lebas, Ali Sié, Catherine E. Oldenburg.

**Software:** Elisabeth A. Gebreegziabher.

**Supervision:** Thomas M. Lietman, Ali Sié, Catherine E. Oldenburg.

**Visualization:** Elisabeth A. Gebreegziabher, David V. Glidden, Benjamin F. Arnold.

**Writing – original draft:** Elisabeth A. Gebreegziabher.

**Writing – review & editing:** Elisabeth A. Gebreegziabher, Mamadou Ouattara, Mamadou Bountogo, Boubacar Coulibaly, Valentin Boudo, Thierry Ouedraogo, Elodie Lebas, Huiyu Hu, Kieran S. O'Brien, Michelle S. Hsiang, David V. Glidden, Benjamin F. Arnold, Thomas M. Lietman, Ali Sié, Catherine E. Oldenburg.

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
