## [Decision Letter · Decision Letter 0]

11 Jul 2025

PGPH-D-25-00813

The role of Seasonal Malaria Chemoprevention in the effect of Azithromycin on Child Mortality: A Secondary Analysis of the CHAT Cluster Randomized Clinical Trial

Dear Dr. Gebreegziabher,

Thank you for submitting your manuscript to PLOS Global Public Health. After careful consideration, we feel that it has merit but does not fully meet PLOS Global Public Health’s publication criteria as it currently stands. Therefore, we invite you to submit a revised version of the manuscript that addresses the points raised during the review process.

The manuscript has been evaluated by two reviewers, and their comments are available below.

Reviewer 2 has raised a number of concerns that need attention. They request additional information on methodological aspects of the study (such as the choice of additive vs multiplicative interactions), and revisions to the statistical analyses to control for underlying assumptions and enhance the impact of some of the analyses presented.

Could you please revise the manuscript to carefully address the concerns raised?

We look forward to receiving your revised manuscript.

Kind regards,

Alejandro Torrado Pacheco, Ph.D.

PLOS Editor

Journal Requirements:

1. Please provide additional details regarding participant consent. In the ethics statement in the Methods and online submission information, please ensure that you have specified (1) whether consent was informed and (2) what type you obtained (for instance, written or verbal, and if verbal, how it was documented and witnessed). If your study included minors, state whether you obtained consent from parents or guardians. If the need for consent was waived by the ethics committee, please include this information.

Additional Editor Comments (if provided):

Reviewers' comments:

Reviewer's Responses to Questions

**Comments to the Author**

1. Does this manuscript meet PLOS Global Public Health’s publication criteria?

Reviewer #1: Yes

Reviewer #2: Yes

2. Has the statistical analysis been performed appropriately and rigorously?

Reviewer #1: Yes

Reviewer #2: No

3. Have the authors made all data underlying the findings in their manuscript fully available (please refer to the Data Availability Statement at the start of the manuscript PDF file)?

Reviewer #1: Yes

Reviewer #2: Yes

4. Is the manuscript presented in an intelligible fashion and written in standard English?

Reviewer #1: Yes

Reviewer #2: Yes

Reviewer #1: This was a post hoc secondary analysis of the CHAT trial and additionally included SMC data from NMCP. The primary outcome for these analyses was all-cause mortality, assessed using a biannual census. The sample size was based on childhood mortality for the overall AZ versus placebo comparison in the CHAT trial. The basic design and analysis plan had been previously reviewed and accepted as valid.

The sensitivity analysis in this renewed or revisited setting accounting for possible appropriate covariates or confounders adjusted for SMC coverage in the analysis by SMC season and they categorized the year into quarters, to evaluate during which time of the year AZ distribution yielded the greatest benefit against mortality. Lastly, they examined the role of SMC coverage in the effect of AZ on child mortality adjusting for age of children and distance to health facility, and their interactions with AZ as both may influence the level of SMC coverage and affect the benefit of AZ for mortality. Apparently in the Results section the sensitivity results did not cause concern.

For appropriate coverage analyses they used Poisson regression models with robust standard errors and similar methods for examining interactions on additive and multiplicate scales. Coverage and mortality investigations are seen in the tables and figures in the text and supplementary materials.

Despite the lack of statistical significance on several fronts, the analysis was done appropriately and the descriptive summary in the discussion was comprehensive giving several limitations, one of which perhaps was the sample size limitation resulting in what the authors describe as lack of power which can happen in a secondary analysis setting.

Reviewer #2: The authors have utilized Poisson regression with person-time at risk as an offset (likely using log(person-time)) to estimate incidence rates and incidence rate ratios. While Poisson regression is appropriate for count-based outcomes, its direct application to mortality data aggregated across time and clusters can lead to overly simplistic interpretations due to assumptions of a constant hazard rate over the observation period. Specifically:

1. Constant Hazard Assumption

The Poisson regression model used by authors implicitly assumes a constant hazard rate (mortality risk) over the duration of follow-up periods. This assumption is questionable given the biologically plausible temporal variability in mortality rates, particularly related to the seasonal effects of malaria and other infectious diseases. Such constant-hazard assumptions typically underestimate effects (particularly if hazards decline or vary significantly over time), potentially reducing power and precision.

Recommendation:

Consider employing survival analysis methods such as the Cox proportional hazards model (multiplicative effects) or other flexible models (e.g., accelerated failure-time models, piecewise exponential additive mixed models). These models naturally accommodate variable hazards over time and can explicitly handle right-censoring and variable follow-up periods inherent in longitudinal cohort data. Cox models, especially with robust standard errors or frailty terms (random effects to account for clustering at the community level), would more appropriately reflect the underlying data structure and temporal variations. The authors' categorization of SMC administration as time periods (January–June vs July–December) could be treated as a time-varying covariate to dynamically model how its effect on mortality changes throughout the study period. Survival models may also show an improved power when using SMC coverage as a continuous covariate.

2. Additive versus Multiplicative Interactions

The authors reported both additive and multiplicative interactions. While reporting both provides thoroughness, it may introduce interpretive complexity. A clearer rationale for choosing additive or multiplicative interactions is beneficial.

Recommendation: Provide explicit epidemiological justification or interpretation for choosing to report interactions on both scales. Consider emphasizing the scale most relevant to public health implications clearly and succinctly in the discussion.

3. Sensitivity Analysis: Dichotomizing SMC Coverage

The authors categorized coverage at an 80% threshold. While using established thresholds is informative, the sensitivity analysis might gain clarity and impact from a more pronounced contrast.

Recommendation:

Perform an additional sensitivity analysis comparing communities in the highest quintile of coverage (top 20%) with the lowest quintile (bottom 20%), providing a stark and potentially more insightful contrast of extremes in SMC coverage.

4. Visual Presentation of Results

Some figures largely duplicates information presented in Tables (eg Fig 5 and Table 3 - Fig 4 and Table 2 - Fig 6 and Table 4).

5. Type I error control

The authors investigated many different interactions with SMC coverage (through time periods, as a continuous measure and also as a dichotomous aggregated variable) with treatment. Looking at both incidence rate differences and incidence rate ratios. While this can look like a thorough exploratory investigation, if one of the methods would yield a significant result, so SMC coverage <80% shows a significant AZ vs Placebo IRD and IRR, with the many tests and comparisons that were performed, could the likelihood of a type I error here be higher?

**Do you want your identity to be public for this peer review?** For information about this choice, including consent withdrawal, please see our Privacy Policy

Reviewer #1: No

Reviewer #2: **Yes: ** Jordache Ramjith

---

## [Decision Letter · Decision Letter 1]

28 Aug 2025

The role of Seasonal Malaria Chemoprevention in the effect of Azithromycin on Child Mortality: A Secondary Analysis of the CHAT Cluster Randomized Clinical Trial

PGPH-D-25-00813R1

Dear Doctor Gebreegziabher,

We are pleased to inform you that your manuscript 'The role of Seasonal Malaria Chemoprevention in the effect of Azithromycin on Child Mortality: A Secondary Analysis of the CHAT Cluster Randomized Clinical Trial' has been provisionally accepted for publication in PLOS Global Public Health.

Best regards,

Alassane Dicko

Academic Editor

The manuscript has been reviewed by two independent reviewers, and their comments have been adequately addressed and at their satisfaction. As newly appointed editor I found the manuscript presents a rigorous secondary analysis of the CHAT trial data on a highly relevant and timely topic. The manuscript is very clear and extremely well written. I congratulate the authors and recommend its rapid publication.

Reviewer Comments (if any, and for reference):

Reviewer's Responses to Questions

**Comments to the Author**

Reviewer #1: All comments have been addressed

Reviewer #2: All comments have been addressed

publication criteria?

Reviewer #1: Yes

Reviewer #2: Yes

3. Has the statistical analysis been performed appropriately and rigorously?

Reviewer #1: Yes

Reviewer #2: Yes

4. Have the authors made all data underlying the findings in their manuscript fully available (please refer to the Data Availability Statement at the start of the manuscript PDF file)?

Reviewer #1: (No Response)

Reviewer #2: Yes

5. Is the manuscript presented in an intelligible fashion and written in standard English?

Reviewer #1: (No Response)

Reviewer #2: Yes

Reviewer #1: (No Response)

Reviewer #2: Thank you for addressing the comments so thoroughly

**Do you want your identity to be public for this peer review?** For information about this choice, including consent withdrawal, please see our Privacy Policy

Reviewer #1: No

Reviewer #2: **Yes: ** Jordache Ramjith
